# Chemical Derivatization in Flow Analysis

**DOI:** 10.3390/molecules27051563

**Published:** 2022-02-26

**Authors:** Fábio R. P. Rocha, Elias A. G. Zagatto

**Affiliations:** Center for Nuclear Energy in Agriculture, University of Sao Paulo, Piracicaba 13416-000, Brazil; frprocha@cena.usp.br

**Keywords:** flow analyzers, green analytical chemistry, UV-Vis spectrophotometry, Schlieren effect, kinetic discrimination, solid-phase reagents, flow programming, manifold programming

## Abstract

Chemical derivatization for improving selectivity and/or sensitivity is a common practice in analytical chemistry. It is particularly attractive in flow analysis in view of its highly reproducible reagent addition(s) and controlled timing. Then, measurements without attaining the steady state, kinetic discrimination, exploitation of unstable reagents and/or products, as well as strategies compliant with Green Analytical Chemistry, have been efficiently exploited. Flow-based chemical derivatization has been accomplished by different approaches, most involving flow and manifold programming. Solid-phase reagents, novel strategies for sample insertion and reagent addition, as well as to increase sample residence time have been also exploited. However, the required alterations in flow rates and/or manifold geometry may lead to spurious signals (e.g., Schlieren effect) resulting in distorted peaks and a noisy/drifty baseline. These anomalies can be circumvented by a proper flow system design. In this review, these aspects are critically discussed mostly in relation to spectrophotometric and luminometric detection.

## 1. Introduction

Modern flow analyzers are highly versatile, allowing the efficient accomplishment of all steps inherent to the analytical procedure, such as sample and reagent handling and flow-through detection [1]. These steps are in-line carried out in an automated manner and expert flow analyzers for managing entire sample batches without human intervention have become a reality [2].

Chemical derivatization is usual in sample processing and a common step in flow analysis. A species with more appropriate characteristics for detection is efficiently produced, thus improving the main analytical figures of merit, especially selectivity, sensitivity, and detection limit. The precise control of reagent addition and timing opens the way for exploiting unstable reagents and products, kinetic discrimination, catalytic reactions, and processes without attaining chemical equilibria. Moreover, chemical derivatization is carried out in a closed environment, thus minimizing risks to the analyst, sample contamination, analyte losses, and side reactions involving atmospheric components. Green Analytical Chemistry (GAC) is efficiently accomplished by saving reagents, minimizing waste generation, and exploiting diverse approaches [3]. The versatility of the flow analyzer for chemical derivatization is expanded by exploiting flow/manifold programming, often involving multicommutation, as emphasized in comprehensive reviews [4,5,6].

The real-time modifications in flow rates and manifold geometry inherent to flow/manifold programming may lead to recorded signal anomalies, such as baseline fluctuations and inverted/noisy peaks of hydrodynamic and/or physicochemical origins. These anomalies may deteriorate the figures of merit, mostly precision and accuracy, and are often circumvented by modifying the manifold design. This review emphasizes the different flow approaches for implementing chemical derivatization, mainly in relation to spectrophotometric and luminometric detection.

## 2. Chemical Derivatization

Chemical derivatization involves the conversion of a compound (educt) into a product with a more favorable chemical structure, the derivative. Already in the 19th century, the approach was exploited in organic synthesis [7]. In general, a specific functional group of an organic compound participates in the derivatization reaction, which converts the educt into a derivative of a different chemical composition, thus modifying reactivity, solubility, boiling/melting points, or the aggregation state. Exploitation of these properties in analytical chemistry may improve the separation, quantification and/or preconcentration efficiencies.

Most initial applications involving chemical derivatization relied on determinations of organic compounds of pharmaceutical relevance by colorimetry or spectrophotometry [8], with or without chromatographic separation. Chemical derivatization is also usual in gas chromatography, aiming the formation of more volatile species, thus expanding the analytical applicability of the technique [9]. In flow analysis, different detection techniques, e.g., atomic spectrometry [10], enthalpimetry [11], amperometry [12], turbidimetry/nephelometry [13], and gravimetry [14], have been associated to chemical derivatization. Nevertheless the process is more usual in relation to spectrophotometry and luminescence [15].

A worldwide consensus on the definition of chemical derivatization has not been attained. Some statements can be, however, mentioned, such as “the conversion of a chemical compound into a derivative (as for identification)” [16]; and “derivatization involves changing in some way the basic chemical or physical structure of a compound, usually to a single product, which may be more useful for the analysis of the original analyte” [17]. Specific definitions are also available, such as “chemical conversion of the substances to be chromatographed into more volatile derivatives” or “derivatization procedures convert non- or weakly luminescent sample molecules to highly luminescent products” for chemical derivatization in gas chromatography [18] and fluorescence [19], respectively. In the present review, all in-line processes (e.g., chemical, photochemical, and electrochemical) yielding a different species to be detected are considered as chemical derivatization. Titrations are not discussed in the present review because the emphasis is not in the derivative, but on the analyte consumption.

### 2.1. Types of Flow-Based Chemical Derivatizations

Although monitoring the derivative of the initial educt is the most usual approach, other alternatives are feasible in flow-based procedures, as discussed below.

#### 2.1.1. Catalytic Methods

The analyte catalyzes an indicator reaction, which yields the species to be monitored, and the analytical signal is usually proportional to the rate of product formation, thus to the analyte content in the assayed sample [20]. These methods are typically characterized by high sensitivity, although selectivity may be a hindrance in some applications because of the catalytic effects of potentially interfering species. Most applications exploit fixed-time methods and flow analysis is very attractive because of the rigorous time control. A pioneering application of a catalytic method in unsegmented flow analysis was focused on vanadium determination at the nanogram level relying on the V(V) effect on the chromotropic acid-bromate indicator reaction [21]. Applications involving the determinations of metal ions, anions, and organic species by different flow modalities, including chip-based analyzers, were recently reviewed [22]. The authors foresaw the exploitation of catalytic methods for post-column chemical derivatization in chromatography.

Catalytic methods are also useful for simultaneous determinations, relying on the distinct effects of each analyte on the rate of the indicator reaction. After successive measurements, each corresponding to a different mean sample residence time, several analytical signals are obtained, each reflecting the different individual analyte contributions. The rigid and reproducible time control inherent to flow analysis is then essential, and a multivariate data treatment can be exploited for the simultaneous determinations [23]. The approach was applied to the catalytic determination of molybdenum in river waters relying on the Mo(VI)-catalyzed iodide oxidation by H_2_O_2_ under acidic conditions [24]. Data were acquired from two sample zones, and multivariate calibration relying on the partial least squares algorithm allowed the simultaneous determination of Fe(III), the main potential interferent.

#### 2.1.2. Photochemical Derivatization

A particular type of chemical derivatization involves photochemical processes, typically carried out under ultraviolet irradiation. In general, the educt modifications rely on either the formation of a radical species or reactions involving radicals, yielding a species which is more efficiently monitored. Fundamentals and applications of photochemical processes in flow analysis, including the exploitation of photo-generated radicals, fractionation and speciation analysis, photo-induced luminescence, and in-line treatment of chemical wastes were reviewed [25]. Recently, photo-derivatization of nitenpyram and pyraclostrobin under UV irradiation combined to optosensing was exploited for the fluorimetric determination of the pesticides in grapes and wines [26].

#### 2.1.3. Electrochemical Derivatization

Analogously to photochemical reactions, electrochemical processes may also yield species more appropriate for detection. The approach involves redox processes under controlled potential at the electrodes, associated to spectrophotometric detection of the products, as demonstrated in the determination of pharmaceuticals in blood plasma [27]. The coulometric generation of reagents for chemical derivatization was also described [28]. Specific aspects of electrochemical derivatization in flow analysis and liquid chromatography were reviewed [29].

#### 2.1.4. Discolorimetry

Some applications rely on the consumption of a monitored reagent by the analyte, and a typical example is discolorimetry. The reagent is a colored species and the reaction with the analyte yields a colorless (or less absorbing) chemical species, so that the absorbance lessening constitutes the measurement basis. In flow-based procedures involving the reagent as the sample carrier or added as a confluent stream, the recorded inverted peak reflects the reagent consumption and constitutes itself the analytical signal. A noteworthy aspect is the possibility of destroying the excess of unstable generated reagent, for yielding a more environmental friendly procedure [30].

Intermittent additions of sample and reagents are also feasible, as exemplified in the determination of acid dissociable cyanide involving the fading of the color of the Cu(I)/2,2-biquinoline 4,4-dicarboxylic acid complex. With tandem streams, the analytical response was measured by considering the transient signals in the absence of cyanide as a reference [31].

#### 2.1.5. Analyte Volatilization

A particular example of chemical derivatization involves the volatilization of the analyte for segregating it from the sample matrix. The analyte is converted to a gaseous species, separated from the sample, and further processed [32]. The chemical species to be monitored seldom mimics the educt. The determination of ammonium exploiting its conversion to ammonia under alkaline conditions, which is transferred to a neutral or acidic stream and directed towards detection is an example [33]. The process was also exploited in the above mentioned determination of acid dissociable cyanide involving HCN diffusion [31]. It is also useful for hydride and cold vapor generation in flow-based atomic spectrometry, sometimes assisted by photochemical processes [34].

A diversity of alternatives is available for gas-liquid separation in flow analysis, including gas diffusion, pervaporation, and membraneless (e.g., isothermal distillation) approaches [35].

#### 2.1.6. Analyte Microextractions

Liquid–liquid and solid–liquid microextractions often involve chemical derivatization of the analytes by a suitable reagent dissolved in the extractant or adsorbed on the solid support. In flow analysis, several microextraction techniques have been exploited and approaches have been proposed to ensure the proper interaction of the species, as discussed in recent review articles [36,37]. Although phase transference seldom reaches equilibrium conditions, precise results are obtained in view of the reproducible sample handing conditions and timing.

Microextractions have been accomplished in segmented flow analysis since the fifties. A pioneering application dealt with the spectrophotometric determination of sulfate in natural waters, based on chemical derivatization (methylthymol blue method) [38]. Sample clean-up was carried out before its inlet into the main analytical channel: the sample flew through a Dowex 50W-X8 cation exchange resin mini-column, where the potential interfering cations (Ca^2+^, Mg^2+^, Al^3+^, and Fe^3+^) were retained, and off-line elution was accomplished from time to time. Up to 7.0 mg L^−1^ Ca^2+^, the main potential interfering species, was tolerated, and a 30 h^−1^ sampling rate was attained.

#### 2.1.7. Other Approaches

The processes involved in the product formation in a chemical derivatization are also of interest in some applications, such as in kinetic analytical methods (item 4.1). Moreover, variations of physicochemical properties, such as the heat release in enthalpimetry [11] and light emission in chemiluminescence [15,39], may also provide useful information.

### 2.2. Derivatization in Liquid Chromatography

Chemical derivatization is often exploited in liquid chromatography, with the aim of reducing matrix interferences, enhancing the analytical selectivity and detectability, as well as improving the analyte separation. It can be off-line or in-line accomplished [40].

Off-line derivatization is time-consuming and laborious as it requires completion of the involved processes, and often large volumes of samples and reagents. These limitations hinder its applicability, especially for large-scale assays, and can be circumvented by exploiting in-line derivatization.

For in-line post-column derivatization, a confluent reagent stream merges with the eluent immediately after the chromatographic column, and the monitorable product is formed inside a reactor just before detection. The involved reactions should be fast, and band broadening due to dilution by the reagent and dispersion inside the reactor needs to be minimized. An advantage is that post-column derivatization is not limited by hydrodynamic pressure, as the eluent stream has already left the chromatographic column.

Pre-column derivatization is beneficial especially regarding compounds difficult to be separated from each other, or compounds that may decompose or react with other components during the chromatographic separation. As it modifies the chemical and physical properties of the educt before the chromatographic separation, these deleterious effects are circumvented. Furthermore, it may improve the analyte-stationary phase interaction. An adverse aspect, however, is that in-line pre-column derivatization may be limited by pressure effects, as it takes place between the sample insertion port and the chromatographic column. To circumvent this limitation, the derivatized educt may fill a sampling loop under ambient pressure, which is thereafter inserted into the pressurized eluent flowing towards the chromatographic column. Exploitation of flow analysis for pre-column derivatization was reviewed [40], and the potential of the lab-on-valve flow analyzer to this end was further discussed [41]. As a recent application, the in-line microwave-assisted analyte derivatization with methyl-*N*-(trimethylsilyl) trifluoroacetamide to yield volatile derivatives for the determinations of atenolol and propranolol in human plasma by GC-MS [42] can be mentioned. Recently, the hyphenation of flow systems with gas or liquid chromatography was pointed out as a trend in flow analysis [43].

A noteworthy situation relies on sequential injection chromatography (SIC), which matches the versatility of flow analysis with the separation ability of chromatography [44]. In-line pre-column derivatization in SIC was pioneering exploited in the fluorimetric determination of intracellular free amino acids in microalgae involving a reaction with o-phthaldialdehyde [45], whereas post-column derivatization was exploited for the determinations of Cu(II), Zn(II), and Fe(II) as 4-(2-pyridylazo)resorcinol complexes [46].

A deeper discussion of this topic is beyond the scope of this review. It should, however, be recalled that chemical derivatization is efficiently carried out by coupling a flow system with the chromatograph.

## 3. Flow-based Approaches for Chemical Derivatization

During the development of flow analysis, a diversity of modalities has been proposed, in part motivated by novel possibilities to implement chemical derivatization. In this regard, the way the reagents are added, the strategies for improving sample/reagents mixing conditions, and the reproducible timings were of utmost relevance.

The state-of-art of flow analysis involves computer-control for implementing all steps of sample processing without the analyst intervention in the manifold architecture. This includes the capability for modifying sample/reagent volumes and flow rates, the addition of different reagents, flow reversal, stream redirecting, flow stopping, zone sampling, etc., and is the essence of flow programming [4], often involving multicommutation [5,6]. These resources have been widely exploited, especially for implementing complex assays, kinetic methods, wide-range analysis, and simultaneous determinations [35].

### 3.1. Strategies for Reagents and Sample Additions

Reagent addition is a cornerstone in flow-based chemical derivatization, as it may affect the analytical figures of merit and the system reliability. Therefore, different possibilities for reagent (and sample) additions into the analytical path have been proposed (Figure 1).

In the first flow injection systems, the sample was inserted into a reagent carrier stream, featuring a single-line flow analyzer [35,47]. This configuration (Figure 1a) is mostly suitable for relatively low sample inserted volumes, because the sample–reagent mixing depends mainly on the axial dispersion. Moreover, for larger inserted sample volumes, longer time intervals are required for the reagent to reach the center of the sample zone, and this may cause peak distortions, such as double peaks [35]. The consumption of the continuously pumped reagent is high, especially for the relatively high flow rates often implemented for improving sample throughput.

Alternatively, the sample aliquot is inserted into a chemically inert carrier stream, which mimics the major components of the sample matrix in order to minimize the Schlieren effect [48], and the reagent additions occur by stream confluences. This confluence configuration (Figure 1b) is inherent to segmented flow analysis [49], and especially attractive in relation to large sample inserted volumes in unsegmented flow analyzers. The continuous reagent additions may, however, lead to a high amount of wasted reagents, which is not consistent with the GAC concepts. A fundamental aspect related to chemical derivatization in confluence flow systems is the sample/reagent volumetric ratio. It depends on the ratios of flow rates of the sample carrier and the confluent streams [50], and may affect the sensitivity and linear response range. When sensitivity is critical, the reagent flow rate is usually set as low as possible to minimize sample dilution at the confluence site. The reagent concentration is then increased accordingly, but this may deteriorate the sample–reagent mixing conditions.

A good practice is to set a not too high sample/reagent volumetric ratio. The sample volumetric fraction in the fluid element associated with the analytical signal undergoes only a reasonable reduction, whereas the improvement in mixing conditions is outstanding. As a rule, sensitivity is improved because the more efficient reagent utilization compensates for the increased sample dilution. Furthermore, the detection limit may undergo a remarkable improvement due to the better mixing. The reduction of the reagent concentration also makes the Schlieren effect less intense [48] and this effect was pointed out as the ultimate aspect restricting the detection limit in flow-based spectrophotometry [35]. It is particularly attractive when the reagent is characterized by low dissolution ability, i.e., mixing of the solutions is relatively slow regardless of the involved solubilities. Another favorable aspect of the confluence configuration is the increase in the sample zone length at the confluence site, which diminishes the post-confluence sample dispersion [50].

A less usual variant of single-line flow analyzers involves injection of the reagent into a continuous flowing sample, featuring a reverse flow injection system, rFIA [51,52]. The resulting configuration (Figure 1c) minimizes the reagent consumption and, under proper conditions, maximizes the amount of the analyte in the sample zone associated with the analytical signal. The approach has been often applied for industrial process control to assess temporal variations in concentrations. To this aim, aliquots of the sample batch are periodically sampled and mixed with the reagent to yield the derivative for detection. With selective reagents, it is possible to monitor the concentration variations of several parameters of interest, such as reagents, products, or by products concentrations.

Reverse flow injection is also a simple alternative for sequential determinations, accomplished by successively injecting different reagents into the flowing sample [53]. The main limitations refer to the high sample consumption and the time intervals for sample replacement and system washing, the latter being critical for large-scale assays. Aiming at system simplicity and ruggedness, the different reagents are usually inserted with a single injector (Figure 1c). Alternatively, system versatility can be expanded by using one injector for each reagent (Figure 1d). Each injection port corresponds to a different analytical path length, thus to a different mean available time for derivatization. The proper choice of the injection port positions is then an additional aspect to be considered in designing the flow analyzer.

Another possibility is to obtain different analytical signals associated to different reaction conditions, as exemplified by the spectrophotometric determinations of silicate and phosphate in river waters. High acidity or addition of oxalic acid plus an increase of sample residence time were exploited to modify the derivatization conditions, thus improving the analytical selectivity of the above-mentioned determinations [54].

Other configurations involve sample and reagent insertions by exploiting different flow modalities [55]. If a single carrier stream is used (Figure 1e), the interaction between sample and reagent zones is restricted to axial dispersion, and the analyzer presents the advantages and limitations inherent to single-line configuration, such as poor sample-reagent mixing, restricted exploitation of high inserted volumes, and a high susceptibility to the Schlieren effect. The approach is typical in sequential injection analysis [56] and related techniques, and the mixing conditions can be improved by exploiting tandem streams [57]. In this later situation, the sample/reagent ratio depends on the number and volume of the multiple inserted plugs, the involved dispersions, and the reagent concentrations.

If the sample and reagent are inserted into convergent carrier streams, a flow system with merging zones [35] is established (Figure 1f), presenting the characteristics of the confluence configuration, although with a significant reduction of the reagent consumption. A perfect overlap of the merging zones can be established by the proper adjustment of flow rates and inserted volumes, thus optimizing the reagent consumption, and characterizing a symmetrical flow system. Alternatively, the flow system can be designed with a partial zone overlap, thus yielding asynchronous systems [58]. The established concentration gradients are then more efficiently exploited and system versatility is improved. Simultaneous determinations involving different conditions for chemical derivatization are then straightforwardly attained.

Solid-phase reagents can also be effectively exploited for chemical derivatization in flow analysis, either as packed mini-columns [59], open tubular reactors [60,61], or suspensions [62,63]. The strategy relies on the reproducible experimental conditions, which ensure a reproducible analyte interaction with the solid reagents. Advantages include a low reagent consumption (only the amount needed for the reaction is consumed), an increase of the reaction rate due to the reagent excess, the possibility of using slightly soluble reagents, the avoidance of dilution effects, and manifold simplification. On the other hand, strategies for minimizing reagent lixiviation (e.g., a reagent chemically bound to the solid support) and backpressure effects (e.g., suitable particle size and/or porous materials) should be applied. Solid samples can be also suitably handled aiming at in-line analyte extraction or sample digestion before analyte derivatization [64].

### 3.2. Mixing Conditions

Suitable mixing is essential for a successful chemical derivatization in flow analysis, especially considering the short sample residence time available. The process relies on diffusion (both axial and radial) and convection, which may be maximized or minimized according to the involved analytical application. To this end, besides the sample and reagent volumes and flow rates, the dimensions and geometry of the reactors (e.g., coiled, knotted, or packed) are relevant [35,47]. Mixing is highly affected by the characteristics of the flowing stream, thus pulsed flows [65], particles fluidization [66], and flow reversal [67] have been exploited to maximize the analyte–reagent interaction. The latter is a typical approach in sequential injection flow analyzers [56].

For improving sample–reagent mixing, a mixing chamber (with or without stirring) is useful, mainly in derivatizations involving several reagents. It is inherent to the flow-batch analyzer [68], which combines the advantages of flow and batchwise systems, being then worthy for increasing sample residence time in derivatizations relying on relatively slow chemical reactions. These features are also efficiently accomplished in the lab-in-syringe flow analyzers, in which chemical processes are carried out inside the plunger of a syringe pump [69].

### 3.3. Reproducible Timing

The reproducible timing characteristic of flow analysis favors the exploitation of processes even when full completion of the involved chemical reactions is not attained. In addition, it is inherent to flow and manifold programming aiming to increase the system versatility, to develop expert flow analyzers [2], and to enhance the productivity of the analytical laboratory. Exploitation of these aspects improves the analytical figures of merit and is a powerful tool to design flow analyzers, outlined as follows.

When the sample residence time in the analytical path should be increased, specific strategies, such as flow stopping, zone trapping, flow rate variations, and zone recycling [35], are worthy. The aim is to minimize the impact of the increased residence time on sample dispersion and sampling rate. In this regard, ingenious strategies involving simultaneous sample processing have been proposed [70].

Flow stopping (Figure 2a) is implemented either by switching the pumping off or by commutation [71], typically when the sample zone reaches the flow cell. Formation or degradation of the product is then efficiently monitored. Zone recycling [35] is a variant of this process: the sample zone is kept flowing in a closed loop, including the flow cell, for detection at different sample residence times. Zone trapping (Figure 2b) relies on the removal of the central part of the sample zone, in which the components are allowed to react further. After a predefined time interval, the zone is reinserted into the carrier, flowing towards detection.

Monosegmented flow analysis [72] (Figure 2c) is an alternative for efficiently implementing relatively slow chemical derivatizations. It is compatible with the different modalities of flow analysis [73], allowing also to accomplish, e.g., liquid–liquid microextractions, flow titrations, and determinations of gaseous species. The sample–reagent zone is sandwiched between two air plugs to restrict dispersion, so that high sample residence times can be efficiently attained. Simultaneous sample processing is also feasible, with beneficial effects on sample throughput. Similar effects can be attained by using air-carrier flow systems [74].

## 4. Highlights of Chemical Derivatization in Flow Analysis

### 4.1. Kinetic Methods

Segmented flow analysis typically involves reactions approaching the steady state, which is less usual in unsegmented flow analysis, where the implemented methods are characterized as fixed-time ones, established by the proper choice of flow-rates and dimensions of the analytical path. The strict and reproducible time control permits also to exploit other kinetic approaches by continuously monitoring the product formation [75]. To this end, the flow is stopped when the sample zone is at the flow cell [76] or otherwise, the central portion of the sample zone is trapped inside a coiled reactor [70]. The former approach is particularly useful for evaluating kinetic parameters and for attaining a more selective analyte determination, including compensation of the radiation absorption by colored samples. Moreover, different approaches for exploiting two residence times for each injected sample, such as sample splitting [77,78], detectors in series [79], relocation of reactors [80], and multi-site detection [81] have been proposed. Analytical determinations without attaining the steady state are typically characterized by high sample throughputs, but the inherent sensitivity lessening may be a hindrance in some applications.

Kinetic discrimination is also feasible, especially when the reaction product is measured without a significant formation of a potential interfering species via a side reaction. The strategy improves selectivity and may be exploited for simultaneous determinations, as exemplified in Table 1 [23,77,78,79,80,81,82,83,84,85,86,87,88,89,90,91,92,93,94,95].

The feasibility of kinetic discrimination in flow analysis was pioneering demonstrated in the determination of magnesium and strontium relying on the differences in the dissociation rate of the corresponding trans-1,2-diaminocyclohexanetetraacetate complexes [96]. The amount of ligand released after two different sample processing times was quantified after complexation with Cu(II) by using two spectrophotometers placed in series, achieving < 9.0% relative errors and a 60 h^−1^ sampling rate. Another classic example is the determination of phosphate by the molybdenum blue method without interference of silicate, which forms an analogous species via a slower reaction [54]. In addition to selectivity improvement, the strategy may be useful for simultaneous determinations of these anions relying on measurements related to two mean sample residence times [54,91]. On the other hand, the analogous batchwise determination of phosphate would require silicate masking.

### 4.2. Unstable Reagents and Products

Another favorable aspect, inherent to the reproducible and controlled timing and reaction conditions, is that highly reactive unstable species may be reproducibly generated in flow systems, paving the way for the exploitation of novel reagents and products [97]. In fact, strong oxidizing, e.g., Ag(II), Co(III), and Mn(III), or reducing, e.g., Cr(II), U(III), and V(II) agents, can be in-line generated from stable species. In this regard, a mini-column filled with the Jones reductor can be used for producing strongly reducing species [98], whereas electrochemical processes are useful for producing strong oxidants [99] to be added to the sample zone. Another example refers to the in-line generation of bromine from bromide and bromate applied to the determination of acetylcysteine in pharmaceuticals [30]. After measurement of the analytical signal, the excess of the reagent was in-line consumed by ascorbic acid aiming at a greener procedure. Applications involving unstable reagents benefit from their generation in a closed system, without contact with the atmosphere. Moreover, the ability to detect unstable yet reproductively formed species allows the proposals of novel analytical methods and represents a clear advantage of flow analysis. Examples of applications are presented in Table 2 [30,100,101,102,103,104,105,106,107,108,109]. Moreover, the instability of the measured species can be exploited to improve selectivity, as demonstrated in the ingenious determination of ascorbic acid relying on its intrinsic UV absorption [110]. After the initial measurement, the sample zone is alkalinized to favor the analyte degradation, thus a second measurement reflects the contribution of the absorption by the sample matrix. The analyte is then selectively determined by difference. The rate of analytes degradation was also exploited for the determinations of carbamate pesticides [83].

Applications involving turbidimetry and chemiluminescence are noteworthy. The former was pioneering exploited for sulfate determination in natural waters based on the reproductive formation of barium sulfate precipitate [111]. Flow-based turbidimetry benefits from the reproducible nucleation and precipitate growing, which tend to be maintained in suspension by the continuous flowing stream [13], especially when pulsed flows are exploited [112]. As chemiluminescence [15] relies on the characteristic radiation emitted during the transient formation of a reaction product or intermediary, fast reagent mixing and strict time control are mandatory to attain reliable results. In fact, the development of flow analysis has stimulated applications involving chemiluminescence [43].

### 4.3. Optosensing

A particular application of solid reagents in flow analysis refers to solid-phase spectrophotometry or optosensing and a broad discussion about this topic is available [113]. The latter term is more general, encompassing other detection techniques, such as molecular luminescence [114]. The reagent is usually immobilized on a suitable solid support placed in the flow-through detector, allowing measurements to be carried out simultaneously with the analyte reaction-retention. The derivative is then retained at the support or otherwise chemical derivatizations may be accomplished directly on the support. The approach is worthy for improving sensitivity as the reaction product is accumulated at the support, and the dilution inherent to analyte elution before measurement is avoided. Selectivity may also be improved due to changes in reactivity of the immobilized reagent or kinetic discrimination [115]. Reversible retention of the analyte and exploitation of the same immobilized reagent for successive measurement cycles is also feasible [116], yielding more environmental friendly procedures. Some applications of optosensing involving chemical derivatizations are highlighted in Table 3 [115,116,117,118,119,120,121,122,123,124].

### 4.4. Simultaneous Determinations and Chemical Speciation

Chemical derivatization is also involved in most simultaneous determinations in flow analysis, either by adjusting the reaction conditions or by adding selective reagents. To this aim, flow and manifold programming [4] have been often exploited. Other strategies involve kinetic discrimination (item 4.1), multi-site detection [81], multi-purpose flow systems [125], asynchronous merging zones [126], sandwich techniques [127], and reverse flow analysis [53], among others.

A noteworthy application relies on chemical speciation, which relies on the differences in reactivity of the species involved in the chemical derivatization. A typical approach involves the selective determination of a species followed by total determination after sample pretreatment involving chemical, electrochemical, or photochemical processes. Classical examples are the determinations of Fe(II)/Fe(III), Cr(III)/Cr(VI), and NO_3_^−^/NO_2_^−^ [128]. The determination of clinical iron parameters (serum iron, unsaturated iron binding capacity, and total iron binding capacity) in human serum, by exploiting sample processing under different acidities and complexation with ferrozine exemplifies a more recent application [129]. Overall, the approach is usually more simple, cost-effective, and faster than the chromatographic ones, although the scope is limited by the number of analytes determined, typically restricted to 2–3 species.

### 4.5. Green Analytical Methods

While in GAC there is a premise to avoid chemical derivatization whenever possible, several approaches have been proposed to minimize its impact in the environmental friendliness, relying mainly on the use of less hazardous chemicals and solvents, and more effective energy sources [130]. In this sense, flow analysis is a powerful tool especially because of its potential to minimize reagent consumption and waste generation. This is inherent to modalities involving the intermittent addition of reagents in the chemical derivatizations, such as sequential injection, multipumping flow, and multisyringe flow analysis, as well as those involving solid-phase reagents. This goal is also successfully attained by miniaturization, including µ-FIA and lab-on-valve. However, the potential of flow analysis to GAC is significantly wider, involving reagentless procedures [131,132], replacement of toxic reagents [133,134], reuse of reagents [116,135], vegetable natural extracts as source of reagents [136] and enzymes [137], as well as in-line waste treatment [134,138]. A wide discussion of this topic is available elsewhere [3].

### 4.6. Expert Systems

The potential of chemical derivatizations in flow analysis is significantly expanded in expert flow systems [2]. This encompasses, for example, in-line optimization of the reactional conditions, in-line adjustment of the reaction medium and flow/manifold programming, aiming, e.g., to avoid matrix effects, and to perform multi-analyte determinations. Accuracy assessment relying on analyte determination by different analytical methods, typically involving different approaches for chemical derivatization, is also feasible [139].

Chemical derivatization in flow analysis has been useful also in relation to wide-range analyte determination, aiming to reduce the number of out-of-scale samples. The sample is processed under different dispersions (e.g., different sample volumes, zone sampling, or both), and further submitted to chemical derivatization. Different analytical curves are then obtained, each corresponding to a given analyte concentration range. Different detectors or derivatization methods characterized by distinct sensitivity or selectivity may also be exploited [2].

## 5. Conclusions

Chemical derivatization is fundamental in flow analysis. By one side, it involves the formation of a better detectable species, leading to selectivity/sensitivity improvement, as well as to advanced flow systems for simultaneous and wide-range determinations. On the other hand, it benefits itself from the inherent characteristics of flow analysis, allowing a better exploitation of chemical reactions without attaining equilibrium, the possibility of kinetic discrimination, and the exploitation of unstable reagents and products, as well as the compliance with the GAC principles.

Chemical derivatization is also linked to the development of several flow modalities, relying on different ways to add and mix the reagents with the sample, as well as to increase the sample residence time without promoting excessive sample dispersion. The proper choice of the flow analyzer modality as well as its design may, therefore, critically affect the analytical performance. In this context, flow programming, multicommutation, and expert flow systems offer several resources for improving the system design and may be considered the state-of-art of chemical derivatization in flow systems.

As final remarks, some reflections on chemical derivatization in flow analysis may be presented:

(i) Most flow-based applications involve a simple mechanization of well-established analytical methods. Nevertheless, the advantages are increased when chemical derivatization exploits the characteristics inherent to flow analysis for, e.g., modifications in flow chemistries, timing, in-line separation-concentration, kinetic discrimination, and simultaneous/wide-range determinations, among others. Exploitation of these aspects is a fertile field for further development, including when a flow analyzer is a front-of-end for chromatography.

(ii) The term “derivatization” is typically associated to chromatography, mass spectrometry, UV-vis spectrophotometry, and luminescence. However, its relevance to flow analysis has been increasingly emphasized, being a positive aspect towards further development. This also holds for µFIA and microfluidic devices.

(iii) In spite of the outstanding current development of flow analysis, an intense human labor is still required to minimize systematic errors and to ensure reliable analytical results and safer working conditions to the analysts. In this context, the development of expert systems plays an important role.

(iv) Several environmentally friendly innovations have been proposed for chemical derivatizations in flow analysis and the flow-based systems paved the way for making chemical derivatization a useful strategy for GAC. This is a counterpoint to the less realistic statement of GAC principle 6 that chemical derivatization should be avoided [140].

## Figures and Tables

**Figure 1 molecules-27-01563-f001:**
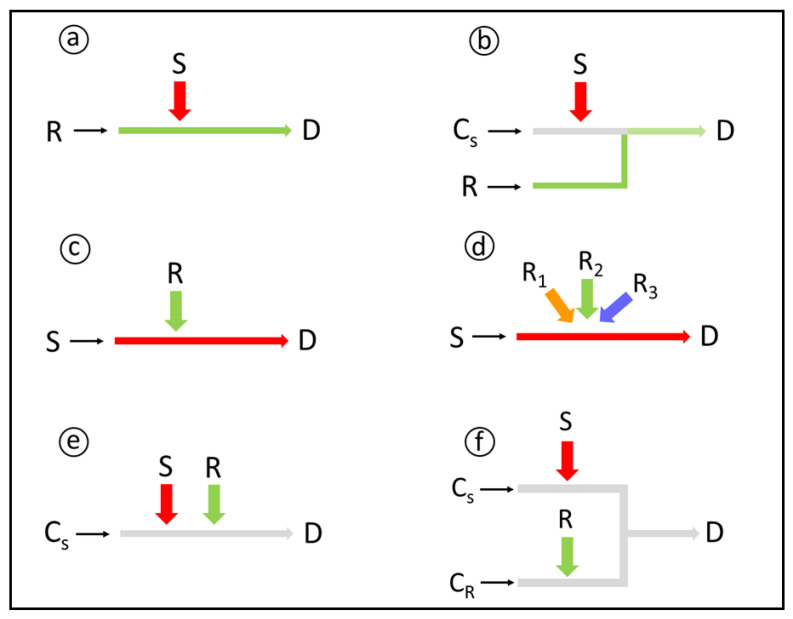
Configurations for sample and reagent insertions in flow analysis. S: sample; R, R_1_, R_2_, R_3_: reagents; C_S_: sample carrier stream; C_R_: reagent carrier stream; D: detection; Large arrows: time-based, loop-based, hydrodynamical or syringe insertions; Small arrows: flow directions; Flow diagrams: **a**—single line, **b**—confluent, **c**,**d**—reverse flow injection, **e**—single line merging zones, **f**—confluent merging zones. Further sample processing may occur before detection. For details, see text.

**Figure 2 molecules-27-01563-f002:**
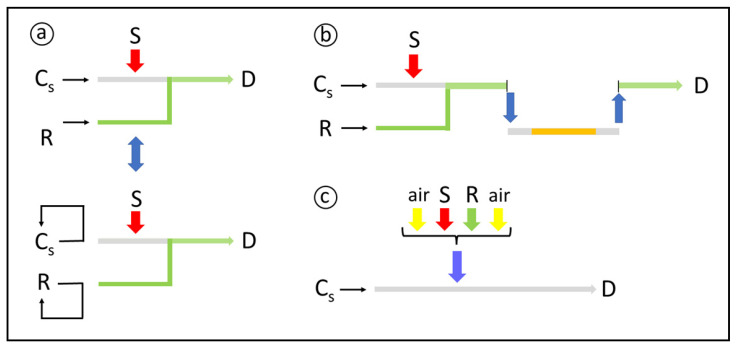
Flow manifolds related to strategies for increasing sample residence time without significantly incrementing sample dispersion. **a**—flow stopping, **b**—zone trapping, and **c**—monosegmented flow analysis. S: sample; R: reagent; C_S_: sample carrier stream; D: detection; Large arrows: time-based, loop-based, hydrodynamical or syringe insertions; Small arrows: flow directions. For details, see text.

**Table 1 molecules-27-01563-t001:** Examples of kinetic discrimination in flow analysis.

Analyte(s)	Sample	Reaction	Sampling Rate (h^−1^)	CV (%)	Remarks	Ref.
Acetaminophen or isoxsuprine, isoniazid	Pharmaceuticals	Reaction with 1-fluoro-2,4-dinitrobenzene releasing fluoride	40	1.8–3.6	Detection by a fluoride ion-selective electrode	[92]
Ascorbic acid, cysteine	Dietary supplements	Reduction of 8-molybdodiphosphate	―	1.3–3.2	Novel approach for data treatment (mean centering of ratio kinetic profiles method)	[93]
Bromate, chlorite	Treated water	Analytes oxide bromide reagent to bromine, which reacts with o-dianisidine	―	8.5–8.8	Sample splitting, reaction at different temperatures	[77]
Bromide	Brine	Bromide oxidation by chloramine T, reaction with phenol red	60	<1.0	Slower reaction with chloride minimising its interference	[94]
Carbofuran, propoxur, metolcarb, fenobucarb	Water, fruits	Hydrolysis/diazotization with p-nitroaniline in alkaline medium	18	0.8–3.3	Data processing by back-propagation/artificial neural network	[95]
Cathecol, resorcinol	Synthetic mixtures	Oxidation by H_2_O_2_ under peroxidase catalysis	60	3.4	Flow stopping associated to multiple linear regression	[82]
Chlorpyrifos, carbaryl	Pesticide formulations	Oxidation by H_2_O_2_ under alkaline medium	80	4.0–6.0	Exploitation of different analytes degradation rates	[83]
Cobalt, nickel	Synthetic mixtures	Complexation with HBAT	―	―	Different strategies to modify the sample residence times	[84]
Cobalt, nickel	Metal alloys	Complexation with PAR from citrate complexes	40	<1.0	Relocatable reactor to achieve two sample residence times	[80]
Copper, nickel	Plant materials	Complexation with Br-PADAP	20	2.0	Relocation of the flow cell for detection at two sample residence times	[81]
Free and total SO_2_	Wines	p-rosaniline method	55	<3.1	Dual flow stopping, measurements before and after alkaline hydrolysis	[85]
Furfural, vanillin	Synthetic mixtures	Reactions with p-aminophenol, yielding Shiff bases	30	0.2–1.9	Zone splitting to achieve two sample residence times	[78]
Gallium, aluminum	Synthetic mixtures	Complexation with PAR	―	0.8–1.6	Flow stopping, principal component regression	[86]
Glucose, fructose	Synthetic mixtures	Analytes oxidation by periodate	―	2.0	Remaning periodate detected by reaction with pyrogallol	[87]
3-Hydroxybutyrate, 3-hydroxyvalerate	Biodegraded polymers	3-hydroxybutyrate dehydrogenase-catalysed reaction with coenzyme NAD^+^	20	0.8–1.5	Exploitation of differential enzimatic reactions with two enzyme reactors and fluorimetric/spectrophotometric detectors placed in series	[79]
Iron, copper	Wastewater, pharmaceuticals	Hydroxylamine oxidation yielding nitrite, determined by Griess method	32–39	1.3–1.6	Microchip with two reaction coils at different temperatures	[88]
Iron, vanadium	Metal alloys	Iodide oxidation by Cr(VI)	50	0.5–3.0	Differential catalitic effect, data treatment by PLS	[23]
Levodopa, benserazide	Pharmaceutical formulations	Analytes oxidation by periodate	20	2.5–4.0	Flow stopping, multiway PLS	[89]
Molybdate, tungstate	Steels	Iodide oxidation by H_2_O_2_	―	1.6–3.4	Mathematical algorithm to compensate the synergistic analytes catalytic effects	[90]
Phosphate, silicate	Waters	Oxidation of thiamine to thiochrome by the molybdate heteropoly acids	60	0.25–0.7	Exploitation of different rates of the molybdate heteropoly acids formation	[91]

Br-PADAP: 2-(5-bromo-2-pyridylazo)-5-(diethylamino)-phenol; HBAT: 2-hydroxybenzaldehyde thiosemicarbazone; NAD^+^: Nicotinamide adenine dinucleotide; PAR: 4-(2-pyridylazo) resorcinol; PLS: Partial least squares regression.

**Table 2 molecules-27-01563-t002:** Illustrative examples of flow-based procedures exploiting unstable derivatives.

Analyte	Unstable Derivative	Application	Ref.
Acetylcysteine	Bromine	Pharmaceuticals	[30]
Bromate, bromide	Intermediate of the 5-Br-PADAP and SCN-reaction	Tap and mineral waters	[100]
Chloride	Chorine produced by photochemical oxidation	Urine and waters	[102]
Cr(VI)	Intermediate of reaction with H_2_O_2_	Fresh and wastewaters	[103]
Cyanide	Intermediate of the pyridine-barbituric acid reaction	Natural and wastewaters	[104]
Ethanol	Ce(IV) ethanolic complex	Alcoholic beverages	[105]
Formaldehyde	Intermediate of reaction with phloroglucinol	Foodstuffs	[106]
Manganese	Intermediate of oxidation of 4,4′-bis(dimethylamino)-diphenylmethane by periodate	Fresh and estuarine waters	[107]
Manganese	Mn(III)/EDTA complex	Freshwaters	[108]
Metoclopramide, tetracaine	Intermediate of pharmaceuticals oxidation by dichromate, in the presence of oxalate under acidic conditions	Pharmaceutical samples	[109]
Total polyphenols	Enol derivative formed by reaction with hypochlorite	Wines, tea	[101]

5-Br-PADAP: 2-(5-dibromo-2-pyridylazo)-5-(diethylamino)phenol; EDTA: Ethylenediaminetetraacetic acid.

**Table 3 molecules-27-01563-t003:** Selected applications of flow-based optosensing involving chemical derivatization.

Analyte	Sample	Derivative	Sorbent	LOD (µg L^−1^)	CV (%)	Remarks	Ref.
Bromate ^a^	Drinking water	Radical formed from oxidation of chlorpromazine	Discovery DSC-MCAX	0.9	<3.6	In-line oxidation of chlor-promazine by the analyte and sorption of the product	[117]
Formaldehyde ^a^	Ethanol fuel	3,5-diacetyl-1,4-dihydrolutidine	C18-bonded silica	30	2.2	In-line derivative formation	[118]
Glucose ^a^	Pharmaceuticals	Thionine	Actigel ALD beads	3 × 10^4^	<5.0	Thionine reduction by NADH yielded in the enzimatic glucose oxidation	[119]
Iron ^a^	Natural waters	Fe(II)/TAN complex	C18-bonded silica	15	4.0	10-fold higher sensitivity in relation to measurements in solution	[116]
Nickel/Zinc ^a^	Copper-alloys	Metal-TAN complexes	C18-bonded silica	―	1.1–2.1	Simultaneous determination by different sorption rates	[115]
Nitenpyram ^b^	Vegetables	Product of analyte photodegradation	Sephadex QAE-A25	0.5	2.6–3.1	On-line analyte photodegradation and transient retention of the derivative at the solid support	[120]
Oxalate ^a^	Vegetables/spicies	Ti(IV)-ECR complex	Silica-titania xerogel	1 × 10^4^	―	Discoloration of the dye adsorbed on the solid support	[121]
Resveratrol ^b^	Beers	2,4,6-trihydroxy-phenanthren	Sephadex QAE-A25	1.0	1.8	On-line analyte photodegradation and transient retention of the derivative at the solid support	[122]
Tin ^a^	Fruit juices	Sn(IV)-Pyrocatechol violet complex	Sephadex QAE-A25	0.3	2.5	In-line derivative formation	[123]
B12 Vitamin ^c^	Pharmaceuticals	4-Aminophthalate	Dowex 1-X8	0.5	5.3	Bead injection of luminol immobilized on the solid support	[124]

^a^: Spectrophotometric, ^b^: fluorimetric, ^c^: chemiluminescence detection. ECR: Eriochrome cyanine R; TAN: 1-(2-thiazolylazo)-2-naphthol.

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
