# Peer review of "Chemical Derivatization in Flow Analysis"

_molecules, 2022, doi:10.3390/molecules27051563_

Round 1

Reviewer 1 Report

The manuscript “Chemical derivatization in flow analysis” by Fábio R P Rocha , Elias A.G. Zagatto, describes the status and the importance of Chemical derivation of flow analysis. The topic is timely
and I recommend the manuscript to be published as a review in the journal
Molecules after the following addition.

1. It would be great if the author can add some pictorial representation of different processes and some schemes as it will be more beneficial for the general audience. 

Author Response

Following your advice, we prepared the Figure 2, which shows the different processes for increasing the sample residence time in flow systems without impairing the sample throughput.

Reviewer 2 Report

The review presents an introduction to the flow analysis process based on chemical derivatization for beginners, however, some advanced applications are also included.  It covers a very wide research field from ways of derivatization and its implementation to green chemistry aspects. In some parts is rather too general (e.g. section 2.1.1 Catalytic methods) while other sections are very specific (e.g. 2.1.3 Electrochemical derivatization).

I have only a few minor points:

  1. The section 4.5 Titrations should be left out. Even though the titrations are based on chemical reactions, they can not be considered derivatization reactions.
  2. In Conclusion, please join points iii) and iv) into one single point.
  3. Emphasize throughout the text which reference is a review article and which is a research paper. It would make the paper easier to follow for readers.
  4. Line 170 – The sentence „A particular approach…“ does not fit here and should be deleted, or moved to another part of the text.  

Author Response

The review presents an introduction to the flow analysis process based on chemical derivatization for beginners, however, some advanced applications are also included.  It covers a very wide research field from ways of derivatization and its implementation to green chemistry aspects. In some parts is rather too general (e.g. section 2.1.1 Catalytic methods) while other sections are very specific (e.g. 2.1.3 Electrochemical derivatization).

Item 2.1.1 was modified and other items in section 2.1 were improved.

I have only a few minor points:

  1. The section 4.5 Titrations should be left out. Even though the titrations are based on chemical reactions, they cannot be considered derivatization reactions.

In agreement with the referee suggestion, item 4.5 was removed. Instead, a brief comment was added in section 2.

  1. In Conclusion, please join points iii) and iv) into one single point.

Thanks for the comment. Item iii) refers to the need for development of expert systems, whereas item iv) discusses the importance of green strategies for chemical derivatization. Accordingly, text was modified to avoid the overlap.

  1. Emphasize throughout the text which reference is a review article and which is a research paper. It would make the paper easier to follow for readers.

The review articles were highlighted in the manuscript.

  1. Line 170 – the sentence „A particular approach…“ does not fit here and should be deleted, or moved to another part of the text.  

The phrase was removed and a brief comment was inserted in section 2.1.3.

Reviewer 3 Report

This review is a clear and well-described contribution about principles and methods developed by chemical derivatization in flow analysis.

There are many self-citations in this manuscript, please comment on the added value of this new review.

Some comments

Complete the title with “with spectrophotometric and luminometric detection” to clearly announce the context to readers.

The sentences in the abtract are to summarize. Too much information in a single sentence led to loss of impact and understanding.

In the abstract at line 6 for Selectivity, Sensitivity and Yield, specify the related analytical chemistry process step.

Line 8, expand on what you mean by "controlled and highly reproducible addition(s) of reagents and timing".

Line 9 explains how "Measurements without reaching steady state, kinetic discrimination, exploitation of unstable reactants and/or products" relate to green chemistry.

In several parts of the manuscript, the authors insist on the green characteristic of this approach. However avoiding derivatization is green. This approach cannot always be considered as green especially with regard to the reagents used, it is necessary to specify in which configuration it is green. And avoids a generalization which is accompanied by a loss of nuance. E.g. Line 37, the issue “save reagents and minimize waste generation” should be considered in the context where it could be considered green. E.g. In general chemical derivarization in flow analysis cannot be claimed as green. E.g. when the sample was inserted into a reagent carrier stream, does not constitute a green approach

The 2.2 part is not on “flow analysis” only since there is chromatographic separation. It must be suppress to focus on flow analysis only, which is the aim of this review.

Line 236, "sample matrix" or "sample solvent"?

For a and b, there is no discussion of the nature of the reagent with regard to detection, which is also a limitation of these approaches.

Line 269

Could you explain why the c disposal it advantageous for industrial process control involving successive analysis of the same sample batch to assess temporal variations in concentrations

In figure1 Ri is not all time used but R, describe more clearly the distinction between R and Ri. And clearly distinguish Cs and Cr. And it seems that in a, c and d there is no streams. And D is detection? Why "towards" since there is arrow on the scheme. For large arrows, you means injection. Globally this figure is important and must be improved in order to be clearer.

Author Response

This review is a clear and well-described contribution about principles and methods developed by chemical derivatization in flow analysis. There are many self-citations in this manuscript, please comment on the added value of this new review.

Most self-citations refer to original proposals, reflecting the pioneering spirit. Anyway, some of them were removed or replaced by more recent ones.

Some comments

Complete the title with “with spectrophotometric and luminometric detection” to clearly announce the context to readers.

We kept the original title in order to avoid an unnecessary particularization. In addition, other detection techniques are considered in the text.

The sentences in the abstract are to summarize. Too much information in a single sentence led to loss of impact and understanding.

We are in full agreement with the reviewer. Thus, long sentences were avoided, not only in the Abstract but throughout the entire manuscript. We believe that readability was improved.

In the abstract at line 6 for Selectivity, Sensitivity and Yield, specify the related analytical chemistry process step.

We believe the sentence is clear, so we did not add any complementation. Anyway, the English language was improved.

Line 8, expand on what you mean by "controlled and highly reproducible addition(s) of reagents and timing".

The phrase was modified for better clarity.

Line 9 explains how "Measurements without reaching steady state, kinetic discrimination, exploitation of unstable reactants and/or products…" relates to green chemistry.

Only “minimized reagent consumption and waste generation” refers to green chemistry. Accordingly, the sentence was modified for better clarity.

In several parts of the manuscript, the authors insist on the green characteristic of this approach. However, avoiding derivatization is green. This approach cannot always be considered as green especially with regard to the reagents used, it is necessary to specify in which configuration it is green. And avoids a generalization which is accompanied by a loss of nuance. E.g. Line 37, the issue “save reagents and minimize waste generation” should be considered in the context where it could be considered green. E.g. In general chemical derivarization in flow analysis cannot be claimed as green. E.g. when the sample was inserted into a reagent carrier stream, does not constitute a green approach.

We agreed that the ideal approach for Green Analytical Chemistry is to avoid chemical derivatization. However, it is not realistic for most applications, especially those involving spectrophotometric and luminometric detection. These aspects were discussed in the manuscript (e.g. item 4.6) and more environmentally friendly strategies were highlighted, whereas other less suitable ones (e.g. single line manifolds, as mentioned by the referee) were pointed out. Anyway, based on the referee comment, the manuscript was carefully revised and some modifications were included for better clarity.

Part 2.2 is not on “flow analysis” only since there is chromatographic separation. It must be suppressed to focus on flow analysis only, which is the aim of this review.

We agreed that this observation is pertinent. However, the flow analysis-liquid chromatography frontier is somewhat confusing, as emphasized in the first book on flow analysis [Ruzicka, J.; Hansen E.H. Flow Injection Analysis, 1st ed.; John Wiley & Sons: New York, USA, 1981. ISBN: 9780471081920] and later by IUPAC [Zagatto, E. A. G.; van Staden, J. F.; Maniasso, N.; Stefan, R. I.; Marshall, G. D. Information essential for characterizing a flow-based analytical system. Recommendations for its use.  Pure Appl. Chem., 2002, 74, 585-592]. Furthermore, liquid chromatography is frequently mentioned in the present review, as some derivatization processes and components are analogous to those of flow analysis. Anyhow, and considering the pertinence of the referee´s opinion, we modified the item 2.2, now a shorter text.

Line 236, "sample matrix" or "sample solvent"?

The term “sample matrix” is correct. To avoid the Schlieren effect, the carrier composition should be as close as possible to that of the sample. The sentence was slightly modified for better understanding.

For 1a and 1b, there is no discussion of the nature of the reagent with regard to detection, which is also a limitation of these approaches.

Unfortunately, the referee comment was unclear to the authors.

Line 269. Could you explain why the 1c disposal it advantageous for industrial process control involving successive analysis of the same sample batch to assess temporal variations in concentrations?

The 1c manifold is relevant as, during an industrial process, the concentrations of some parameters undergo variations. Information on this aspect can be a subsidy for controlling the entire process. Text was amended to include this information.

In figure1 Ri is not all time used but R, describe more clearly the distinction between R and Ri. And clearly distinguish Cs and Cr. And it seems that in a, c and d there is no streams. And D is detection? Why "towards" since there is arrow on the scheme. For large arrows, you mean injection. Globally this figure is important and must be improved in order to be clearer.

Thanks for the comment. Figure 1 and caption were modified for better clarity.